# Immunometabolic Signature during Respiratory Viral Infection: A Potential Target for Host-Directed Therapies

**DOI:** 10.3390/v15020525

**Published:** 2023-02-13

**Authors:** Larissa Menezes dos Reis, Marcelo Rodrigues Berçot, Bianca Gazieri Castelucci, Ana Julia Estumano Martins, Gisele Castro, Pedro M. Moraes-Vieira

**Affiliations:** 1Laboratory of Immunometabolism, Department of Genetics, Evolution, Microbiology and Immunology, University of Campinas, Campinas 13083-862, SP, Brazil; 2Department of Immunology, Institute of Biomedical Sciences, University of São Paulo, São Paulo 05508-270, SP, Brazil; 3Graduate Program in Genetics and Molecular Biology, Institute of Biology, University of Campinas, Campinas 13083-970, SP, Brazil; 4Experimental Medicine Research Cluster (EMRC), University of Campinas, Campinas 13083-872, SP, Brazil; 5Obesity and Comorbidities Research Center (OCRC), University of Campinas, Campinas 13083-872, SP, Brazil

**Keywords:** respiratory viruses, immunometabolism, host-directed therapies, RNA viruses, immune responses, inflammation

## Abstract

RNA viruses are known to induce a wide variety of respiratory tract illnesses, from simple colds to the latest coronavirus pandemic, causing effects on public health and the economy worldwide. Influenza virus (IV), parainfluenza virus (PIV), metapneumovirus (MPV), respiratory syncytial virus (RSV), rhinovirus (RhV), and coronavirus (CoV) are some of the most notable RNA viruses. Despite efforts, due to the high mutation rate, there are still no effective and scalable treatments that accompany the rapid emergence of new diseases associated with respiratory RNA viruses. Host-directed therapies have been applied to combat RNA virus infections by interfering with host cell factors that enhance the ability of immune cells to respond against those pathogens. The reprogramming of immune cell metabolism has recently emerged as a central mechanism in orchestrated immunity against respiratory viruses. Therefore, understanding the metabolic signature of immune cells during virus infection may be a promising tool for developing host-directed therapies. In this review, we revisit recent findings on the immunometabolic modulation in response to infection and discuss how these metabolic pathways may be used as targets for new therapies to combat illnesses caused by respiratory RNA viruses.

## 1. Introduction

Respiratory RNA viruses induce a wide variety of illnesses that affect the respiratory tract, including the nose, mouth, throat, breathing passages, and lungs [1]. These diseases show different degrees of severity and infants, immunocompromised, and elderly people are at higher risk of developing severe diseases. The high potential respiratory RNA viruses have to cause diseases is associated with their capacity to spread rapidly through the air, infecting many people from different locations in a short period. Respiratory viruses were responsible for the main epidemics in recent years, such as SARS in 2003 [2] and H1N1 in 2009 [3]. Recently, the world has witnessed the onset and continuation of a pandemic caused by the SARS-CoV-2 coronavirus, leading to the deaths of millions of people [4,5]. Epidemics and pandemics affect public health and the economy worldwide and are associated with increased hunger and exacerbated social inequality in the world [6,7]. Therefore, understanding the mechanisms of propagation and infection of these viruses is the main strategy to prevent future pandemics.

There are six main types of respiratory RNA viruses: influenza virus (IV), parainfluenza virus (PIV), metapneumovirus (MPV), respiratory syncytial virus (RSV), rhinovirus (RhV), and coronavirus (CoV) [8]. Influenza is a genetically diverse group of viruses of the family *Orthomyxoviridae* and the most common cause of respiratory illness, from seasonal flu to endemic and pandemic events, such as avian H5N1 influenza [9]. Parainfluenza and metapneumovirus belong to the *Paramyoviridae* virus family and are predominant in cases of lower respiratory tract illnesses during childhood, although reinfection in adult life with milder symptoms is common [9,10]. The respiratory syncytial virus belongs to the family *Pneumoviridae*, which causes severe lower respiratory illness mainly in young children and elderly people [11]. Rhinovirus belongs to the *Picornaviridae* virus family and is responsible for the seasonal common cold [12]. Currently, coronavirus, a *Coronaviridae* virus, is the most famous respiratory virus due to the biggest pandemic of the century, in addition to their role in severe diseases such as severe acute respiratory syndrome (SARS) and Middle East respiratory syndrome (MERS) [13].

Viruses are obligate intracellular parasites, modulating host cellular machinery to favor their replication and propagation, displaying a variety of ways to evade the immune response, battling against the defense mechanism of the host (Figure 1). Respiratory viruses infect ciliated or non-ciliated epithelial cells of the host airway tract, the first barrier to infection [14]. These cells have mechanisms to detect pathogens with pattern recognition receptors (PRRs), responsible for recognizing pathogen-associated molecular patterns (PAMPs) or other intracellular components exclusive to invaders, such as nucleic acids and proteins [14,15]. After detecting pathogens through PRRs, these receptors trigger a signaling cascade that limits the replication and spread of viruses due to the induction of interferons, specifically type I and III IFNs, as well as other proinflammatory mediators, such as cytokines and chemokines [16]. The presence of pathogens and accumulation of inflammatory factors recruit and activate innate immune cells, which occur rapidly and in a nonspecific way [17]. This immune mechanism is limited in effectiveness but provides the first defense against infections while recognizing components of pathogens and leading to the activation of adaptive immunity, a response that is highly specific and effective [17]. T lymphocytes, adaptive immune cells, recognize specific major histocompatibility complex (MHC) molecules presented by antigen-presenting cells (APCs), resulting in T lymphocyte activation, maturation, and proliferation by clonal expansion [18]. B lymphocytes start to produce antibodies and T cells migrate to the infection site to kill infected host cells. After infection resolution, there are T and B memory lymphocytes responsible for a faster and more effective response against a second encounter with the same pathogen [19].

Immune cell activation is the main defense mechanism of the host against respiratory viral infection. However, these diseases present different degrees of severity that are frequently associated with exaggerated immune action [20]. Cytokine storm, a consequence of uncontrolled immune response activation, is observed in severe COVID-19 [21,22], influenza [23], and other viral infections [24]. Vaccination is currently the most promising therapeutic strategy against viral infections, generating controlled activation of the immune system and preparing immune response intensity during a real infection. However, respiratory viruses are RNA viruses, presenting a high mutation rate and consequent high genetic variability, which increases the likelihood of vaccine-induced immunity escape and requires frequent vaccine updating [25].

Host-directed therapies are an emerging concept that diverts the focus from pathogens, which require unique targeted treatments, and directs efforts towards what is occurring in the host. These therapeutic strategies include commonly used drugs, immunomodulatory agents, and cellular therapy [26]. HDT is a promising strategy for viral infections, interfering with pathways that are necessary for viral propagation and balancing the immune response by activating the response against invaders and reducing exaggerated response and its effects on disease severity [27].

Upon infection, immune cells remodel their metabolism to perform their effector functions [28]. Immunometabolic reprogramming is associated with both the successful response against invaders and the uncontrolled immune response to infection [29]. Therefore, immunometabolism has proved to be an interesting target in the development of HDT [30,31,32,33,34]. In this review, we will revisit recent findings on immunometabolic modulation in response to infection and discuss how these metabolic pathways may be used as targets for new therapies to combat illnesses caused by respiratory RNA viruses.

## 2. Innate Immune Response: The Front Line in the Fight against Infection

The innate immune system is a rapid and nonspecific response to respiratory viral infections. Respiratory viruses produce an inflammatory microenvironment that promotes the innate immune response, recruiting and activating effector cells as a frontline against the invaders, such as antigen-presenting cells (dendritic cells), phagocytes (macrophages and neutrophils), killing cells (natural killer cells), and granulocytes (basophils and eosinophils).

### 2.1. Dendritic Cells

Dendritic cells (DCs) are critical to the normal functioning of the immune system, having an essential role in monitoring mucosal surfaces and initiating and modulating the adaptative immune response to environmental cues [35,36]. There are two major subtypes of DCs in the lung: migratory/conventional DCs (cDCs) such as CD103+ (cCD1) or CD11b+ (cCD2), efficient in antigen recognition and presentation, and plasmacytoid DCs (pDCs), which are major sources of IFN-I [37,38,39]. In the absence of antigens, cDCs are the most common DC in the lung, responsible for mucosal surveillance and adaptative response modulation [40]. These cells develop in bone marrow (BM) and migrate to peripheral tissues as precursors in a process dependent on CC-chemokine receptor type 2 (CCR2) and CX3 chemokine receptor 1 (CX3CR1) stimulation [38,41,42,43]. Once in the tissues, precursors can uptake antigens by the engagement of pattern recognition receptors (such as the Toll-like receptors, TLR) or be activated by epithelial cytokines, initiating their maturation program [36,40]. The maturing cDCs increase the expression of CC-chemokine receptor type 7 (CCR7), allowing them to migrate to the local draining lymph nodes (dLNs) towards a CCL19/CCL21 gradient [44]. In the dLNs, cDCs’ antigen presentation to naïve CD8+ and CD4+ T cells via major histocompatibility complex class I (MHC I) and class II (MHC II) molecules leads to T cell activation and initiation of the adaptive immune response to those local cues [45].

During respiratory viral infections, pDCs develop a similar response, taking up and processing viral antigens and efficiently presenting them to CD8+ and CD4+ T cells, helping to orchestrate an adequate immune response against the pathogens [40]. Although pDCs are much lower in frequency than cDCs in the airway at a steady state, these cells are rapidly recruited from the bone-marrow-derived monocytes to the lungs after a pathogenic insult via CCR5 [46] and CCR2 [47]. Moreover, pDCs have a less prominent function in presenting antigens to T or B cells and have high expression of TLRs 7 and 9, as well as retinoic acid-inducible gene I (RIG-I) and melanoma differentiation-associated protein 5 (MDA5), which allows them to perform a crucial antiviral response by producing high levels of type I and III interferons (IFN-I and -III) and, to a lesser extent, IL-12 [48].

During respiratory viral infections, the activation and migration of cDCs are necessary for the induction of antiviral responses of CD8 T cells. Under the infection of highly pathogenic viruses, such as severe acute respiratory syndrome coronavirus 2 (SARS-CoV-2) or pneumotropic influenza viruses (such as the H1N1 and H5N1 influenza virus strains), a failure of CD8 T activation, which is necessary for viral clearance, can result in uncontrolled inflammation, leading to tissue injury [49]. During the first 24 h post-infection with influenza virus A (IAV), lung cDCs, activated by the phagocytic engulfment of virions or infected epithelial cells, extensively migrate to dLNs in a CCR7-dependent manner [50]. Upon migration to dLNs, cDCs cross-present viral antigens to naive T cells, inducing T cell activation and an initial cytotoxic response. At initial times, cDC1 cells, which have a superior ability to cross-present antigens, are responsible for most CD8 T cell stimulation during the acute infection. However, after 5 days post-infection, there is an increase in cDC2 migration to the dLNs, which become the dominant subset that stimulates CD8 T cells, necessary for the generation of long-term protection [51,52,53]. Although this division of labor between the cDC subtypes occurs in IAV infection, both in respiratory syncytial virus (RSV) and coronavirus (SARS-CoV, SARS-CoV-2, and MERS-CoV) infections, cDC subtypes migrate simultaneously to the lymph nodes and induce a cytotoxic response [54,55]. Moreover, although associated with an acute inflammatory response and intense monocyte/macrophage activation [56], coronavirus infections impair cDC function, resulting in a defective antigen-specific T cell response [57,58]. Diminished T cell activation leads to poor disease outcomes in SARS-CoV and SARS-CoV-2, which was demonstrated to be worsened in aged individuals [59] and associated with high expression of PGD2 receptor 1 (DP1) by cDCs in mice [60,61].

Besides the accumulation of pDCs in the lung and dLNs upon both RSV and IAV infection and their capacity to produce type I interferons, these cells have a critical role only in RSV infections [62]. Although the depletion of pDCs leads to enhanced virus titers, inflammation, and airway hyperresponsiveness in RSV-infected lungs [62,63], pDC depletion does not affect the clearance of the IAV from the lung [50]. Additionally, in models of IAV infection, pDC responses can also be detrimental, being responsible for enhanced mortality during infection with a lethal dose and possibly associated with excessive production of type I IFNs, which results in uncontrolled inflammation and apoptosis of bronchial epithelium [64,65]. Studies have shown that the responsiveness of pDCs is vital for controlling SARS-CoV-2 infection and COVID-19 severity [66]. It is well described that SARS-CoV viruses are bad inducers of IFN I activation [67,68,69], resulting in elevated inflammatory cytokine levels, vascular leakage, and impaired virus-specific T cells. The decreased IFN I activation by SARS is associated with two significant factors: (1) As in RSV, SARS-CoV developed evasion strategies to escape the IFN I signaling pathway [70]; (2) Although SARS-CoV-2 is more susceptible to IFN I than other coronaviruses, this virus impairs DC activation, resulting in fewer pDCs and a decreased production of IFN I and III [71,72]. Additionally, the early administration of IFN I in mice infected with SARS-CoV attenuates the immunopathology of the disease [73]. Therefore, lung DCs play an essential role during the host response to respiratory viral infections, not only by mounting the cytotoxic antiviral response necessary for viral clearance but also by controlling the degree of inflammation, contributing to diminished tissue damage and disease severity.

As detailed above, viral infections change DCs’ state of activation and function, resulting in profound modifications in their metabolic profiles. When activated by distinct TLR or in response to acute infection, cDCs switch their metabolic program from mitochondrial OXPHOS fueled by the β-oxidation of lipids to aerobic glycolysis (also known as the Warburg effect), decreasing mitochondrial activity and respiration [74,75]. Glycolytic activation of cDCs can be divided into two major time points. In the early-phase induction, phosphatidylinositol 3-kinase (PI3K)/Akt signaling in association with TANK-binding kinase 1 (TBK1) and IκB kinase ε (IKKε), which allows the translocation of the glycolytic enzyme HK2 to the mitochondria, triggers the upregulation of costimulatory molecules, cytokine production, and enhancement of T cell stimulatory capacity [74,76,77,78]. The first phase of glycolytic induction occurs in an mTORC1- and HIF1α-independent manner and can be inhibited by the activation of adenosine monophosphate-activated protein kinase (AMPK) and also by IL-10 [75]. Nevertheless, the late phase in glycolytic induction is tightly regulated by the mammalian target of rapamycin complex 1 (mTORC1), which boosts HIF1α expression and suppresses OXPHOS, potentially due to increased production of nitric oxide by inducible nitric oxide synthase (iNOS) and nitric oxide [79]. Although treatments of DCs with 2DG, a glycolysis inhibitor, prevent DC activation by TLR4 [74], the diminishment of glycolytic metabolism by iNOS inhibition on DCs does not change their ability to upregulate costimulatory molecules, showing that late glycolysis may be dispensable in the DC maturation/activation program [79].

Recently, it was demonstrated that type I IFN production after RIG-I signaling is distinctly affected by glycolysis inhibition in cDCs (monocyte-derived DCs (moDCs)) and pDCs [80], resulting in the blockage of I IFN release by cDCs without changing pDC secretion, which is supported by OXPHOS metabolism. Activation of distinct metabolic profiles in pDCs also depends on the type of receptors activated. Whereas HIF-1α induces an increased glycolytic metabolism after pDC stimulation of TLR7 by specific respiratory viruses such as influenza and RV-16 virus or the synthetic TLR7 agonist gardiquimod [81], some evidence has pointed out that the activation of pDCs by TLR9 stimulation increases not only glycolysis but also OXPHOS and FAO [82]. Additionally, studies have pointed out that FAO of de novo synthesized fatty acids may also be essential for IFNα, TNFα, and IL-6 production by pDCs [82,83], showing that inhibition of FAO can limit pDCs expression of CD86 (a co-stimulatory receptor ligand), resulting in compromised T cell activation [82]. However, as observed in cDCs, mTOR activation seems necessary for pDCs’ production of cytokines, such as IFNα, IL-6, and IL-10 [84,85,86,87]. In conclusion, the authors indicated that the differences observed in cDC and pDC metabolic adaptations seem to be linked to the distinct viral sensor repertoire once pDCs rely on endosomal TLRs and RIG-I signaling. In contrast, cDCs engage mainly in TLR activation to respond to viral infections, using glycolysis as a source of energy and molecules necessary for biosynthesis and the induction of antiviral machinery [74].

### 2.2. Macrophages

Macrophages are innate immune myeloid cells and professional phagocytes originating from two different sources: one that originated in the yolk sac and fetal liver and seeds tissue macrophages and constitutes the so-called tissue-resident macrophages, and one that is dependent on hematopoietic stem cells and monocyte migration, called monocyte-derived infiltrating macrophages [88]. Tissue-resident macrophages occupy different niches and anatomical locations with distinct functional phenotypes: microglia in the central nervous system (CNS), osteoclast in bones, Kupfer cells in the liver, histocytes in the spleen, and even lymphoid organs can have different macrophage phenotypes [89]. In the lungs, macrophages can be divided into three subgroups: alveolar macrophages (AM), interstitial macrophages (IM), and intravascular macrophages [90]. Once in the tissue, macrophages are responsible for immune surveillance, inflammatory regulation, and tissue homeostasis [91], and under steady-state conditions are referred to as naïve macrophages (or M0). In brief, when incited by an inflammatory stimulus, macrophages can undergo different polarization states, often referred to as pro-inflammatory macrophages (or M1-like) and anti-inflammatory macrophages (or M2-like), although it is well-known that macrophage phenotypes comprise a constellation of spectrums [92]. This dichotomy is based on the presence of T cells producing IFN-γ or IL-4, depending on the amount of cytokines and time of exposure, tilting macrophages towards M1 or M2 polarization, respectively [93]. Activating macrophages with lipopolysaccharide (LPS) and IFN-γ results in the profound reshaping of cell metabolism: flux through glycolysis and the glycerol–phosphate and malate–aspartate shuttle increase, which supports energy metabolism and cytoplasmatic NADH production [94], while OXPHOS is downregulated to support the production of pro-inflammatory metabolites and redox signals, such as TNF-α, IL-1α, IL-1β, IL-6, IL-12, and IL-23 [95]. On the other hand, macrophages stimulated with IL-4 induce a different transcriptional program associated with mitochondrial biogenesis and rely mainly on OXPHOS for ATP generation and cellular metabolism [96]. It is worth mentioning that these polarization states do not reflect the actual spectrum of macrophage activation and polarization. If provoked, inflammatory monocytes (Ly6hi) are rapidly recruited from the blood by cytokines released by inflammatory macrophages (CCL2, CCL3, CCL4, and CCL5), differentiate in macrophages and dendritic cells (DCs), and work together with tissue-resident macrophages to resolve inflammation and promote tissue repair [92]. However, the evolution of adaptive immunity enabled macrophages with functions that influence both T and B cell responses: along with the professional antigen presentation cells (APCs) DCs, they serve as a key group of professional APCs and regulate adaptive immunity [97].

Under a steady state, alveolar macrophages account for 90–95% of the cellular content of alveoli; one alveolar macrophage is detected in approximately three alveoli, and macrophages are thought to migrate to other alveoli through Kohn pores [98]. Alveolar macrophages are distinct from macrophages from the airway epithelium and blood vessels, which indicates a greater degree of macrophage specialization and compartmentalization in the lungs [99]. IMs are found in the parenchymal space and interact with DCs and lymphocytes. Interstitial macrophages are found in the parenchymal space and interact with DCs and lymphocytes. Lung macrophages can be further separated into CD11c^hi^Siglec-F^hi^CD11b^lo^MHCII^lo^ alveolar macrophages and CD11c^int^Siglec-F−CD11b^hi^MHCII^hi^ interstitial macrophages [97]. In an ex vivo experiment, alveolar macrophages stimulated with IFN-γ expressed CD69, Toll-like receptor 2 (TLR2), TLR4, CXC-chemokine ligand 9 (CXCL9), CXCL10, CXCL11, and CC-chemokine ligand 5 (CCL5), whereas AM alveolar macrophages stimulated with IL-4 expressed CD206, matrix metalloproteinase 2 (MMP2), MMP7, MMP9, the tyrosine-protein kinase MER, growth arrest-specific protein 7 (GAS7), CD163, stabilin 1 (STAB1), arginase, and A3 receptor [100]. Alveolar macrophages suppress the immune response, inhibiting the DC-mediated activation of T cells and producing TGF-β and retinoic acid, which induces the generation of regulatory T cells Foxp3+ Treg cells, while interstitial macrophages are thought to promote immunity by presenting antigens to interstitial T cells [97,101]. This specific immune tolerance process occurs only in the lungs and not in lymph nodes and depends on the macrophage- and not DC-mediated antigen presentation. On the other hand, swapping the tolerogenic mode to the inflammatory mode goes along with increased secretion of IL-1, IL-6, TNF, and TGF-β [97]. The prevention of inflammatory macrophages is also mediated by different receptors expressed in alveolar epithelial cells (AECs), alveolar fluid, and macrophages, including CD200R, TREM2, TGFβR, SIRPα, IL-10R, Mannose Receptor, TREM1, TREM2, GM-CSF, and PPARγ [99,102,103,104,105,106]. There is still no consensus on whether lung macrophages under steady-state conditions are M1-like or M2-like: exposure to cigarette smoking increases the population of IL-13-producing macrophages [100], suggesting that healthy lungs predominantly have an M1-like polarization phenotype. However, other studies have shown that 50% of human alveolar macrophages express CD206+ [107], which indicates that alveolar macrophages do not fit under the M1/M2 classification. Importantly, M2-like alveolar macrophages are described to participate in allergic diseases in mouse lung tissue [108]. Macrophages express diverse types of pathogen-recognition receptors (i.e., TLR family, C-type lectin, NLPR family) that are activated during infections [109]. Alveolar macrophages constitute the first line of defense: they are connected to AECs by connexin Cx43 hemichannels, which dock each other, sampling pathogens that may enter the alveolar lumen transported by the alveolar liquid flow [98]. However, alveolar macrophages contribute not only to the inflammatory process but also to repairing damaged tissue and resolving inflammation by limiting and restoring normalcy after tissue injury [97]. During inflammation or infection, the collectin Surfactant Protein A/D (SP-A, SP-D) and C1q promote phagocytosis by binding to pathogens or apoptotic cells with their globular heads, and their tail to CD91 in alveolar macrophages [110]. Intriguingly, depending on the milieu, SP-A and SP-D can have anti/pro-inflammatory effects that are mediated through the tail or the head of surfactants and the interaction with macrophage receptors [111]. 

The lung airspace microenvironment is dynamic and has marked environmental fluctuations with different particles being presented to alveolar macrophages, which are responsible not only for “housekeeping” functions, i.e., clearing cell debris, but also for mounting a robust inflammatory response to pathogenic agents. It is suggested that damaged AECs and consequently loss of ligands expressed by these cells lead to macrophage activation and a pro-inflammatory profile [106]. Whether these cells are apoptotic or necrotic seems to be important for the macrophage anti/pro-inflammatory response: on the one hand, apoptotic cells require anti-inflammatory alveolar macrophages to suppress the inflammatory response against self-proteins, and activation of SOCS1 and SOCS3 via the TAM family inhibits cytokine production and TLR signaling, and on the other hand, necrotic cells release damaged-associated molecules that promote inflammatory macrophage activation [106,107,112].

At a steady state, interstitial macrophages are considered “nonalveolar” macrophages, almost absent in the airway lumen [112], located in the lung tissue, and not associated with blood vessels [113]. Interstitial macrophages can be further separated in SiglecF−CD11c+/−CD11b+CCR2+/−CX3CR1+, and a fraction of mouse interstitial macrophage IMs that express Mertk+CD64+CD11b+SiglecF− also express CD11c and MHC-II, like cDCs, but differ from DCs by a high expression of CD64, Mertk, and F4/80 [112,114]. Whereas alveolar macrophages can self-maintain with minimal contribution from circulating monocytes, with a turnover rate of 40% in 1 year [115], interstitial macrophages are, at least in part, replenished from circulating monocytes from different tissues [116]. Interestingly, studies have suggested the possibility of different three interstitial macrophages within the lung, with one subset displaying a higher turnover rate than the two others [108,114]. Different from alveolar macrophages, AMs that originate from yolk-sac-derived monocyte precursors cells, a part of the interstitial macrophages that are seeded to the lungs during embryonic development and constitute “primitive” interstitial macrophages, are located at submesothelial and perivascular locations in adults, and a second wave that rapidly develops from bone marrow gives rise to “definitive” interstitial macrophages located in the lung parenchyma [116]. Interstitial macrophages, like alveolar macrophages, are phagocytic cells and could be considered as a second line of defense, but their most-studied properties arise from their immunoregulatory capacity, which mostly relies on IL-10 production both in mice and humans [117,118]. As the lung microenvironment is dynamic and constantly exposed to immunostimulatory molecules, interstitial macrophages seem to contribute to lung homeostasis by changing lung cDC functions via IL-10 secretion, controlling an allergic-specific T helper type 2 (Th2) response [112].

During infection, viruses (i.e., influenza) replicate in lung epithelial cells and also in macrophages, but infected macrophages do not replicate and, in doing so, do not release infectious progeny [119]. Macrophage infection is a hallmark of early virus recognition by the innate immune system. As macrophages are the first line of defense during infections, and because of their diversity and distribution in the body, several viruses have evolved to infect and replicate inside differentiated macrophages, monocytes, and their precursors [120]. While sampling different pathogens and molecules, macrophages are skewed into different functional phenotypes, interacting with viral pathogens and antimicrobial responses. Not surprisingly, different viruses can affect macrophage polarization and cause immunosuppression or immunopathology accompanied by viral persistence and/or co-infections [121]. Mostly highly pathogenic viruses cause severe pathology by eliciting an M1-associate inflammation, which in some cases can promote a cytokine storm, inducing viral spreading by increasing the lymphocyte flux of infected monocytic cells and causing macrophage death through direct infection [122]. Virus-mediated macrophage death, as described for SARS (and COVID-19), pandemic influenza, African swine fever virus (ASFV), and porcine reproductive and respiratory syndrome virus (PRRSV) [58,120,122,123,124], leads to a series of pathological consequences associated with macrophage polarization by diminishing the antiviral defense performed by M1 macrophages, diminishing secondary antiviral signaling, causing tissue damage, and inducing an M2 polarization state before viral clearance, which viruses use to form a persistent infection and retard resolution [58,120,122,123]. One of the best-studied examples of respiratory viral infection is influenza. Influenza is airborne and first affects the upper respiratory tract. Macrophages sense the influenza A virus in infected cells via pattern-recognition receptors (PRR), a type of TLR, retinoic acid-inducible gene-1 (RIG-1), and the NOD-like receptor NLRP3, which recognize viral RNA, the key PAMP of influenza A [121]. Signaling via these receptors leads to the production of proinflammatory cytokines and type I interferons [125]. The virus can spread to the lower tract, causing lytic inflammation and damaging the blood–tissue barrier, yielding macrophages, NK cells, neutrophils, and later virus-specific T lymphocyte influx [125]. Macrophage depletion following influenza infection is associated with an increased CD8+ T cell response, and the depletion of macrophages after 48 h led to an impaired CD8+ T cell response [126,127]. Macrophages deficient in GM-CSF or its receptor GM-CSFR result reduced viral clearance and higher mortality [128,129]. During infection, macrophages protect the host by phagocytosing surfactant phospholipids and apoptotic cells that accumulate in the alveoli and impair O_2_ and CO_2_ exchange, and mice infected with influenza virus and lacking alveolar macrophages die from hypoxia and lung failure [97]. Most importantly, macrophages can limit lung injury caused by proinflammatory responses without affecting the efficiency of the T cell and B cell responses [127].

COVID-19 disease is characterized by the secretion of numerous cytokines, such as IL-1α, IL-1β, IL-6, IL-7, TNF, type I and III IFN, CCL2, CCL3, and CXCL10 [130], which can lead to a systemic inflammatory process called cytokine storm, which is historically also described for influenza-like syndrome [131]. This influenza-like syndrome takes place once the host immune system is overstimulated by heightened cytokine expression produced via recognition of DAMPS released from epithelial cells or via PAMPS (i.e., TLR-2 and TLR-4) and RIG-I [130]. A common feature of viral infection is the high metabolic demand required to maintain rapid viral replication, which they carry out by reprogramming host metabolism and impeding immune defense [132]. These perturbations in cell metabolism can directly impact host intracellular metabolite levels and dysregulation of metabolic-related enzymes that may ultimately affect the cellular immune response. Citrate and succinate are two metabolic intermediates of the TCA cycle and can exert different functions on innate immune cells: citrate accumulates in TLR-4-activated macrophages and promotes the production of prostaglandins, NO, and ROS [133], whereas succinate has been recognized as a signal that induces IL-1β in a HIF-1α-dependent manner [95,134,135].

Glycolysis and glycolytic rates are also key players in the immune response. Elevated aerobic glycolysis can lead to increased lipid biosynthesis, and non-esterified fatty acids (NEFA) have been reported to increase cytokine production of TNF-α, IL-1β, IL-6, and IL-10, which may aggravate the inflammatory state [136]. Metabolic enzymes, like RNA-binding proteins, such as GAPDH, can bind to TNF-α mRNA, decreasing its expression and yielding glycolysis [137]. In a paper published by our group, we showed that elevated glucose levels favor SARS-CoV-2 replication and proinflammatory cytokine expression, increasing ROS production, and consequently stabilizing HIF-1α, enhancing glycolytic-related gene expression and promoting the cytokine-storm, epithelial cell death, and T cell exhaustion [22]. Moreover, PPARs promote JAK-mediated phosphorylation of STAT proteins, yielding IFN signaling [136].

In the core metabolism of carbohydrates and lipids are mitochondria, which need to reshape their metabolism upon viral infection. To do so, they constantly undergo fission and fusion processes to eliminate damaged mitochondria [138]. mtROS activates mitochondrial antiviral-signaling proteins (MAVS) to produce IFNβ, eliciting the antiviral response [139]. Mitochondria also serve as platforms for MAVS to assemble signaling complexes downstream of RIG-I and activation of NF-κB [140]. Influenza A proteins can translocate to the mitochondrial inner membrane through Tom40, inducing mitochondrial fission and impairing the RIG-I-mediated interferon response [141], and mitofusin 2 (MFN2) can interact with MAVS and impair activation of NF-κB downstream of RIG-I [142,143].

### 2.3. Microglia

As part of the innate immune response, microglia are resident mononuclear phagocytes in the central nervous system (CNS) and participate in pathogen recognition, initiation, and maintenance of local immune responses [144]. Under homeostatic conditions, microglia contribute to the maintenance of brain plasticity, such as synapse pruning and clearance of dead cells [145,146]. Additionally, microglia also scan the brain parenchyma, detecting the occurrence of pathologies [147].

In 1918 the world witnessed one of the deadliest pandemics in human history, the “Spanish Flu”, as it came to be known. During the following years a curious phenomenon, yet to be fully elucidated, began to emerge: a global, decade-long encephalitis lethargica pandemic [148]. It is clear now that different viruses, e.g., influenza virus, can infect the CNS and cause neurological disorders [149]. However, with the current COVID-19 pandemic, much more attention has been given to viruses that are known to be non-neurotropic, that is, their primary site of infection is not the brain, but yet are “neuropathogenic”, which may lead to neurological abnormalities and psychiatric disorders [148,150,151,152]. Interestingly, some speculations about some kind of viral etiology behind neurological disorders such as Parkinson’s disease, acute disseminated encephalomyelitis, and multiple sclerosis have gained attention as coronaviruses, such as HCoV-229E and HCoV-OC43, have been detected in patients’ brains [153,154].

An interesting question arises on how these non-neurotropic viruses, which most of the time are respiratory viruses, may enter and disrupt the CNS. There are currently two routes described for viruses’ entry into the CNS: the hematogenous route, which is characterized by infiltration of peripheral infected cells (such as monocytes/macrophages) into the CNS, crossing the blood–brain barrier; and the neuronal/axonal route, which includes different peripheric nerves such as the olfactory sensory neurons, utilized for retrograde and anterograde movement during neuronal transport [155,156].

Microglia contribute to host survival during the acute phase of viral infection in the CNS through DAMPs recognition, which activates intracellular signaling cascades, resulting in the expression of proinflammatory and antiviral cytokines [157]. However, during viral infection, e.g., COVID-19, the viral replication in lung epithelial cells can impair alveolar O_2_/CO_2_ exchange, leading to systemic hypoxia, including the CNS, which can activate aerobic glycolysis and stabilize HIF-1α in microglia [158]. Indeed, activation of microglia to a pro-inflammatory profile is associated with enhanced aerobic glycolysis for ATP generation, increased expression of the glucose transporter GLUT1, and hexokinase-2 to ROS and NO production, as their generation depends on NADPH, which is a substrate for both the NADPH oxidase (NOX), which produces ROS, and inducible NO synthase (iNOS), which produces NO [158]. CNS hypoxia can lead to a series of disorders: cerebral vasodilation, brain swelling, interstitial edema, headaches, and, in severe cases, degraded brain function, bulbar conjunctival edema, coma, and/or death [154]. 

Systemic cytokine release during respiratory viral infection may lead to important consequences in the brain due to indirect activation or priming of microglia [159,160]. It has been shown that microglia reactivity and pro-inflammatory cytokine release, i.e., IL-1β, IL-6, TNF-α, and IFN-α, match the same time point in which cognitive impairments are observed in mice [160]. This activation also increases mRNA expression of CD36 and CD68 and decreases the expression of the pre-synaptic glutamate transporter VGLUT1, yielding a subtle imbalance in glutamatergic synapse transmission [159]. Of note, conditions that yield a proinflammatory systemic state, such as diabetes and obesity, may damage the blood–brain barrier and may activate and prime tissue-resident immune cells, including microglia [161]. Primed microglia are linked to excessive proinflammatory microglia activation upon viral infection, not only leading to neurological disorders and cognitive disabilities but also having consequences in serotonergic and noradrenergic synapses [148,158].

Different classes of viruses apply distinctive mechanisms to metabolically reprogram host cells in ways that are advantageous for viral replication [22,162]. A growing body of literature indicates that, similarly to macrophages, microglia switch their metabolic status according to their function [163]. However, contrary to macrophages, which are known to produce and respond to TCA cycle metabolic intermediates, i.e., itaconate and succinate, as described above, little is known about their immunomodulatory effects in microglia.

### 2.4. Neutrophils

Neutrophils, also called polymorphonuclear leukocytes, are the most abundant innate immune cell type in the blood [164]. These cells are generated inside the bone marrow through the progressive maturation of progenitor cells [165]. At a steady state, immature neutrophils remain in the bone marrow controlled by CXCR4 upregulation, a receptor internalized by constitutive expression of CXCL12 by bone marrow stromal cells [166,167,168]. After infection stimulation, mature neutrophil numbers increase about ten times [164]. Neutrophils are activated and released from the bone marrow through signaling mediated by type I cytokines, such as GC-CSF, G-CSF, and IFNγ, that downregulate CXCR4 and upregulate genes of the CC receptor family [166,168,169,170]. 

Neutrophils present high mobility and are exposed to diverse immunological environments presenting a plastic metabolism that allows their adaptation to adverse conditions [171]. These cells use carbohydrates, proteins, lipids, or amino acids for energy production, depending on fuel availability in the microenvironment and their stage of differentiation and activation [172]. Glycolysis is the main energy resource of quiescent neutrophils [173]. Neutrophil progenitors reside in the bone marrow and present low bioenergetic activity. The hypoxic microenvironment of the bone marrow induces a glycolytic metabolism linked to HIF-1α stabilization [174]. During differentiation, immature neutrophils present abundant mitochondria and shift their metabolism from glycolysis to fatty acid oxidation [175,176]. Riffelmacher and colleagues showed that autophagy-mediated lipid degradation provides free fatty acids to support mitochondrial respiration to produce ATP [176]. Mature neutrophils are released into the circulation, returning to a quiescent state, but are also highly dependent on glycolysis in an HIF-1α-dependent manner [177] with reduced mitochondria number [175]. 

Neutrophil activation depends on autocrine purinergic signaling triggered by ATP generated by mitochondrial metabolism and released into the extracellular space to facilitate adjacent cell communications [178,179]. An example is the P2Y2 purinergic receptor, which is stimulated by mitochondrial-released ATP and promotes mTOR signaling, which amplifies mitochondrial activity in neighboring cells by a feedback loop and enhances neutrophil migration [180]. Santos and colleagues demonstrate that glutamine administration diminished the number of neutrophils and the levels of CXCL12 in the inflamed lungs, suggesting that chemotaxis is impaired by exogenous glutamine [181].

Activated neutrophils present different functions that are sustained mainly by glucose metabolism [182]. Inflammatory neutrophils present more glycogen storage and accumulation of lipid droplets than blood neutrophils [183,184]. Under normoxia conditions, neutrophil-mediated resolution of the inflammatory responses is controlled by Phd2 expression, a HIF-prolyl hydroxylase enzyme. Sadiku and colleagues demonstrated that loss of Phd2 induces an exaggerated neutrophil response through increased glycolysis and glycogen storage [185]. In inflamed sites, where oxygen and glucose are limited, neutrophils rely on glycogenolysis to generate energy [184,186,187]. Mitochondrial functions are required to sustain neutrophil functions but are dispensable for the rapid initiation of the response to infections [188]. During infection, glucose diverts from glycolysis to PPP to generate NADPH and induces NADPH oxidase for ROS generation, essential to neutrophil extracellular trap (NET) formation [189,190] and respiratory bursts [191]. Glutamine also induces ROS production via NADPH oxidase [192].

Neutrophils play diverse functions against viral infection, reducing viral replication and spread, and activating adaptive immune responses [193]. At a steady state, resident lung neutrophils, called lung-marginated neutrophil poll, are found in the pulmonary vasculature and perivascular space [194,195]. However, their functions are not well-understood [196]. Neutrophils are also detected in the respiratory tract, including the lungs and bronchoalveolar lavage after respiratory viral infection [197], such as infection with metapneumovirus [198], respiratory syncytial virus [199], influenza [200], and coronavirus [201]. During viral infection, virus PAMPs induce an inflammatory response after recognition by pattern-recognition receptors (PRRs) [196]. Neutrophils transmigrate to the site of inflammation after their recruitment by neutrophil chemoattractants produced in the lungs and airways, including CXCL1, CXCL2, and IL-17 [194,199,202]. Some studies suggest that in respiratory-virus- induced inflammation, neutrophil recruitment is associated with a worsened disease, exacerbating local inflammation and tissue damage [196,200,201]. Habibi and colleagues demonstrate that volunteers destined to develop symptomatic infection by nasal administration of respiratory syncytial virus presented inflammatory signals mediated by neutrophils and IL-17 response suppression [203].

The inflammatory microenvironment may induce metabolic reprogramming of neutrophils associated with severe outcomes of respiratory virus diseases [184,204,205,206,207,208]. The respiratory syncytial virus is associated with the production by neutrophils of oxygen radicals through arachidonic acid metabolism responsible for tissue damage and bronchoconstriction during pathogenesis [209,210]. Hypoxia and low glucose availability, features of hyper-inflamed and damaged lung sites affected by a viral infection, induce neutrophils to shift their energy source to protein breakdown by mTORC1 inhibition [211].

Neutrophils are altered in severe COVID-19 [212,213,214]. Poor prognosis in COVID-19 patients is associated with altered glycolytic metabolism that causes elevated chemotaxis of monocytes and neutrophils [215] and leads to the overproduction of ROS via NADPH oxidase [216]. Hypoxia leads to HIF-1α stabilization and neutrophil metabolism adaptation by increasing glycolysis and glycogenolysis for NET formation [214].

### 2.5. Natural Killer

Natural killer (NK) cells belong to the group of lymphoid cells, performing the interface between the innate and adaptative immune response [217], identified as CD3-CD56+ cells, representing 5–15% of human peripheral blood mononuclear cells (PBMCs) and constituting about 10% of lymphocytes in the healthy human lung [218]. NKs are cytotoxic effector cells [219], activated by cytokines such as IL-12, IL-15, and IL-18 [220] with functions ranging from cytokine production, mainly IFNγ and TNFα, to apoptosis induction via death receptor ligands (e.g., FasL), and cell death by the release of cytotoxic granules containing granzymes and perforin [219,221].

Metabolic rearrangements have been associated with NK activation, involving enzyme upregulation, nutrient uptake, and mitochondrial mass increase [222]. Glycolysis and oxidative phosphorylation increased in a Srebp-dependent manner are necessary for IFNγ production and cytotoxic function [223]. Glucose is necessary for both glycolysis and OXPHOS [224]. Amino acids, such as glutamine, are essential to cMyc regulation, a transcription factor involved in the IL-2/IL-12 function responses, but glutaminolysis appears not to be involved [225]. Finally, mTORC1 is robustly regulated in activated NK cells, upregulating glucose uptake and glycolysis, processes necessary for IFNγ and granzyme B production [226].

During acute viral infections, NK cells reprogram their metabolism to execute their antiviral functions. HIF1α controls the metabolic adaptation of NK during virus infection [227]. Salomon and colleagues [222] demonstrate that a robust metabolic response occurs at the peak of the NK response, involving elevated rates of glycolysis and OXPHOS. NK cells enhance glycerolipid and inositol phosphate metabolism to enhance cytotoxicity during SARS-CoV-2 infection [206].

### 2.6. Eosinophils 

Eosinophils are immune cells involved in different sets of cellular processes and are well appreciated for their role in parasitic infection. Beyond this, eosinophils play a key role in diseases such as asthma, eosinophilic gastrointestinal disorders, and systemic hypereosinophilic diseases [228]. Eosinophils express different receptors for a range of cytokines, chemokines, and adhesion molecules, yielding their participation in inflammatory responses and homeostasis [229]. They produce different granule proteins, including major basic proteins (MBPs) 1 and 2, eosinophil cationic protein (ECP), eosinophil peroxidase (EPX, also known as EPO), eosinophil-derived neurotoxin (EDN, also known as RNase2), and cytokines [230]. Upon infection, eosinophils are recruited to the lungs and degranulate into the lung parenchyma, and epithelial cells secrete different eosinophil chemoattractants [231] and may contribute to adaptative immunity by producing EDN, CCL17, CCLL22, CSC-chemokine ligand 9 (CXCL9), and CXCL10, and also serve as antigen-presenting cells [232]. 

Eosinophils seem to play a role in respiratory infections, as fibroblasts and epithelial cells in culture respond to RSV by secreting MIP-1-α and IL-8, which are associated with degranulated EDN and ECP [231], and are known to express an array of TLRs, TLR-7 being the most abundant [233]. When activated, eosinophils produce large amounts of nicotinamide adenine dinucleotide phosphate oxidase (NOX2)-dependent ROS, which may suggest that O_2_ consumption in activated eosinophils is mainly for ROS production and not for OXPHOS ATP generation [234]. Upon activation, IL-3, IL-5, and GM-CSF increase glycolysis, glutaminolysis, lactate production, and mitochondrial oxygen consumption, which sustain OXPHOS and produce ROS [235]. Eosinophils are activated via the TLR-7-Myd88 pathway dependent on IRF-7, IFN-β, and NOS-2 [236], accompanied by the production of NO by iNOS due to TLR-7 activation, with the last being recognized as the most important event for eosinophil-mediated viral clearance [237].

### 2.7. Complement System

The complement system is formed by a complex network of proteins organized in a proteolytic cascade to enhance the function of antibodies and phagocytic cells in response to infections. Three different ways can activate the complement system: the classical, lectin, and alternative pathways [238]. 

Classical activation, known as antibody-dependent, occurs by the recognition of antigen–antibody complexes formed by the C1q protein linked to the surface proteins of pathogens and the Fc region of the IgG and IgM protein isotypes [238]. The lectin pathway is activated by highly conserved carbohydrate structures of pathogens, called pathogen-associated molecular patterns (PAMPs), that are recognized by recognition pattern receptors (PRRs) [238]. Finally, the alternative pathway is spontaneously activated by hydrolysis and remains basally active, which allows a rapid response during pathogen infection [238]. The three pathways converge to the C3 convertase production, which leads to the generation of classical inflammation by anaphylatoxins, the membrane attack complex (MAC), and opsonin, the effectors of the complement system. The functions of the complement system consist of neutralization of pathogens by opsonization, lysis and death of pathogens and infected host cells by the formation of MAC, regulation of the inflammatory response, and enhancement of the adaptive immune response [238,239]. 

The complement system acts as a functional bridge between the innate and adaptive immune system [240]. Classical activation of the complement system depends on the neutralization of pathogen-derived antigens by B-cell-produced antibodies [239]. In addition, the complement system induces DC maturation and consequently presentation of foreign pathogens and activation of T cells [241]. The complement receptor CD21 modulates B cell functions (such as antigen internalization and presentation) and the complement regulator CD46 regulates T-cell-mediated immunity [240].

Viral infection activates the complement system by three activation pathways, leading to viral neutralization by the production of proinflammatory anaphylatoxins and the formation of the MAC to lyse enveloped viruses or infected host cells [242]. Viral antigens processed by the complement system also activate the adaptive response [243], enhancing Th1 response [244] and B cell memory viability [245] and modulating Treg and Th17 responses [246]. The antiviral response triggered by the complement system is essential in the neutralization of respiratory viruses; however, recent data have shown that its hyperactivation is behind severe cases of SARS-CoV-2 [247,248,249,250] and influenza [251,252] virus infection.

In addition to its direct role in innate and adaptive immune response activation, the complement system also plays a role in the modulation of immunometabolism [253]. Liszewski and colleagues [254] demonstrated that CD46 acts as an intracellular nutrient sensor and upregulates the glucose transporter GLUT1 and the aminoacid transporter LAT1, fueling T cell activation [254]. CD46 and TCR costimulation induces mitochondrial production of ROS, which stimulates the secretion of bioactive IL-1β and promotes T helper 1 differentiation [255].

## 3. Adaptive Immune Response: The Better Weapon against Infection

The adaptive response occurs through the interaction between innate immune cells (mainly DCs and macrophages) and the adaptative immune response, the T and B cells. cDCs carrying virus-derived antigens migrate to lymphoid tissue and are presented to CD4 T cells in the lymph nodes via MHC-II, activating a program of differentiation and activation, promoting the migration to the edge of the lymphoid follicle and proliferation. Furthermore, CD8 T cells migrate to the infection site to exert their cytotoxic activity, while B cells activate and produce antibodies, coordinating virus-infected cell elimination. The adaptive immune system is more specific and effective, although slower to initiate. This response comprises the second line of defense against pathogens, formed by humoral and cellular immunity, mediated by virus-specific antibodies produced by B cells and cytotoxic response by T cells, respectively [256].

### 3.1. T Lymphocytes

The adaptive immune system comprises the second line of defense against pathogens, formed by humoral and cellular immunity and mediated by virus-specific antibodies and T cells, respectively [256]. T cell metabolism has been under intense investigation because of its potential therapeutic implications. Upon viral infection, CD4+ T cells, CD8+ T cells, and regulatory T cells (Tregs) are induced [257]. Like activation and differentiation of lymphocytes that lead to metabolic shifts, viral infections have been shown to reshape cell metabolism in ways that are central to the antiviral immune response. CD4+ T cells are activated after recognizing virus-derived MHC class II-associated peptides on APCs that also express co-stimulatory molecules [127], whereas CD8+ T cells are activated in the lymph nodes and recruited to the site of infection, eliminating infected cells via their cytolytic activity [258]. 

Stimulation through the antigen receptor and co-stimulation via CD28 increases glucose transporter GLUT1 expression, glucose uptake and glycolysis, and mitochondrial capacity [259]. However, the metabolic reprogramming of T cells into different polarization states (during cell differentiation and/or infection) is not shared [260]. T effector cells are known to induce aerobic glycolysis upon viral infection, whereas memory T cells rely on mitochondrial metabolism and lipid oxidation, but still can rapidly shift to glycolysis [261,262]. On the other hand, regulatory T cells do not rely on glycolysis or glutamine uptake and instead prioritize mitochondrial lipid, pyruvate, and lactate oxidation [263,264,265,266]. Tregs can be highly glycolytic but their primary transcription factor Foxp3 is capable of repressing glycolysis, and high glycolytic rates have been shown to impair Tregs’ suppressive capacity [264,267]. 

Upon viral infection, RLRs induce mitochondrial antiviral signaling protein (MAVS) and Interferon regulatory factor 3 (IRF3), and MAVS is attached to the mitochondrial membrane [257]. When RLR is activated, MAVS interaction with hexokinase 2 (HK2) is abolished, in turn reducing the glycolytic rate and diminishing intermediates downstream of HK2, and the lactate produced binds to MAVS and suppresses type I interferon expression [268], which may be a reason why viral infections induce high glycolytic rates. Infection may induce host metabolic changes and affect pathogenesis, and the most relevant from a clinical view is catabolic wasting of energy stores [257]. This phenomenon is associated with the loss of lipid stores in adipocytes and the modulation of key regulators of lipid metabolism. Interestingly, reducing type I interferon signaling and CD8+ T cells reduces the severity of whole-body wasting, and CD8+ T cells are suggested to promote weight loss upon viral activation [269]. Beyond regulating polarization and effector functions, cellular metabolism also controls antigen presentation. T cell recognition of peptides is dependent on MHC I and MHC II. MHC I can recognize viral proteins and present them to CD8+ T cells, causing apoptosis and cell death in infected cells [270]. Aconitate Decarboxylase 1 (ACOD1) is an enzyme responsible for the production of Itaconate, a TCA intermediate. IFN-β-induced ACOD1 expression enhances MHC I function via the expression of transporter proteins associated with antigen presentation [271].

### 3.2. B Lymphocytes

B lymphocytes are the antibody (Ab)-producing cells, mediating the humoral immune response [272]. Their activation occurs after encountering specific epitopes that are recognized by surface Ig in an APC- and T-cell-dependent manner. APCs present the small pathogen-associated peptide fragments with the major MHC molecules that bind to TCR on the surface of T cells. A second binding event involves MHC and CD4 molecules, causing costimulation, which leads to additional activation signals within T cells and their CD40 ligand (CD40L) expression. This surface molecule binds to CD40 on B cells, resulting in an activation cascade mediated by cytokine secretion that binds to fully activate B cells, yielding Ab production [273]. B cells producing Ab play a critical role in protection mediated by vaccination [274]. These cells protect against respiratory viral infections, such as influenza [275], respiratory syncytial virus [276], and coronavirus [277]. On the other hand, dysregulated B cells are associated with severe cases of respiratory diseases [278,279,280,281].

Metabolic reprogramming occurs in B cells during their differentiation and activation. Naïve B cells have increased glucose uptake, oxygen consumption, and lactate secretion [282]. B cell activation and cytokine production enhance the glycolytic flux [282,283], while flux through the pentose phosphate pathway (PPP) is limited [284]. Mitochondrial activity may inflect in apoptosis [285].

HIF activity is high in pre-B cells and decreases at the immature B cell stage. HIF reduction is necessary for normal B cell development and their genetic diversity activation [286]. Hypoxemia, common in severe cases of COVID-19, is related to dysfunctional B cells with increased HIF expression [287].

### 3.3. Memory Lymphocytes

Memory T cells are generated during the first antigen contact and are activated on a second encounter. Their rapid expansion and cytotoxic characteristics enable a rapid and efficient response to pathogen control [18]. Their function is related to the success of vaccination-mediated protection [274].

Naïve T cells are metabolically quiescent. However, they become highly active after activation, increasing their demand for substrates. Fatty acid oxidation is the main substrate to fuel OXPHOS in memory T cells [262]. Fatty acids are produced by the cell itself [288] or captured directly from the microenvironment [289].

## 4. Immunometabolism and Host-Directed Therapies for Respiratory Viruses

Viruses, especially RNA respiratory viruses, present a high mutation rate that may contribute to the frequent occurrence of vaccine-induced immunity escape and requires frequent vaccine updates [24]. Host-directed interventions based on the immune response have shown promise in the treatment of respiratory viral infections, but these strategies sometimes come with unpleasant side effects. Cytokine treatment (e.g., type I interferons) increases the immune response; however, it may be associated with exaggerated responses, leading to severe cases of respiratory diseases, such as cytokine storm [22,23] and dysregulation of neutrophils [200,201] and B cells [287]. Despite this, immunosuppressants (e.g., corticosteroids) suppress the immune response, preventing exaggerated inflammation, but they may benefit viral replication and propagation [290].

Immunometabolism reprogramming is associated with the regulation of the immune response during viral infection (Figure 2). Understanding the metabolic regulation of cells during the immune response can contribute to the development of host-directed interventions for pathogen infections [290]. Immunometabolism has been demonstrated as a target for host-directed therapies for fungal diseases [34] and tuberculosis [291,292]. Recently, therapeutic interventions for respiratory viral infections focused on immunometabolism have also been proposed. Some drugs targeting immunometabolism regulation have shown promising effects in the control of severe cases of viral infections (Table 1).

Dimethyl fumarate and 4-octyl-itaconate, derivatives of itaconate, are anti-inflammatory drugs that inhibit NLRP3 inflammasomes [293]. Liao and colleagues [294] demonstrated that the anti-inflammatory effects of 4-octyl-itaconate are also associated with glycolysis reduction via GAPDH inhibition. These compounds attenuate the airway inflammatory response to SARS-CoV-2 virus infection [295] and suppress pulmonary inflammation and mortality in influenza infections [296].

Metformin, an antidiabetic drug that inhibits mTOR and actives AMPK, promotes antifibrotic effects in the lungs [297,298]. Kamyshnyi and colleagues demonstrated that, during SARS-CoV-2 infection, AMPK activation by metformin activates lymphocyte subpopulations, reducing hyperinflammation and controlling the cytokine storm [299]. The PI3K/AKT/mTOR pathway coordinates metabolic reprogramming (i.e., uptake and utilization of nutrients, including glucose and glutamine) and activation of immune cells [300]. Inhibition of PI3K/mTOR reverted the disbalance of glucose metabolism in severe cases of pediatric influenza [301].

Hypoxemia is a defining feature of acute distress syndrome (ARDS), associated with complications of pulmonary inflammation [302,303]. Viral replication in lung epithelial cells impairs gas exchange, leading to hypoxemia, which stabilizes HIF1α in immune cells and reprograms their metabolism. HIF-1α induces heightened rates of glycolysis to produce lactate and ATP, diverting glycolysis to PPP-dependent NADPH and ROS production [304]. HIF-1α stabilization is necessary for immune responses against pathogens but is also associated with an exaggerated inflammatory response by inducing a cytokine storm [22] and elevated chemotaxis of monocytes and neutrophils to the infection site [215]. Hypoxia can dysregulate the response of macrophages [22], neutrophils [214], and lymphocytes [287]. Hypoxia altered the bone marrow hematopoiesis with consequences for the phenotype and number of monocytes and led to the persistence of inflammation [208]. Treatment with mCSF-1 (macrophage growth factor colony-stimulating factor 1) drove inflammation resolution [208]. Understanding the hypoxia and HIF1 α modulation in both mild and severe infections by respiratory viruses may allow the development of new therapeutic approaches for the most severe cases of respiratory syndromes.

Lung injuries are triggered in severe cases of viral infection by an exaggerated inflammatory response [305]. PPARy, a nuclear receptor essential to lipid metabolism, is involved in the shift from inflammatory to anti-inflammatory immune cell polarization, modulating neutrophil function and macrophage activation [306]. 15-Deoxy-Δ12,14-prostaglandin-j2, a PPARy agonist, reduces the number of immune infiltrates and induces a resolving immune response, reducing exaggerated inflammation [307,308,309].

**Table 1 viruses-15-00525-t001:** Metabolic targets and their inhibitor/agonist compounds to immunometabolism modulation during respiratory viral infection.

Intervention	Metabolic Target	Proposed Biological Function	Immune Cell	Respiratory Viruses
DMF/4-Octyl-Itaconate	GAPDH inhibition	Anti-inflammatory effects [294]	T cells [310] B cells [311] NKs [312] DCs [313] Monocytes/Macrophages [295,296] Microglia [314] Neutrophils [315]	SARS-CoV-2 [295] Influenza [296]
Metformin/AICAR	AMPK agonist	Anti-inflammatory effects [316]	T cells [299] B cells [317] Macrophage [318] Monocytes [319] Neutrophils [320]	SARS-CoV-2 [299]
Rapamycin	PI3K/mTOR inhibition.	Immunosuppressive effects [321]	T cells [321] Macrophage [321,322] NK [323] DCs [82]	SARS-CoV-2 [324] Rhinovirus [325] Influenza [326] Respiratory syncytial virus [327]
LY294002	PI3K/AKT inhibition	Immunosuppressive effects [321]	T cells [328] Macrophage [329,330] Dendritic cells [331] Neutrophils [332] NKs [333]	Influenza [334] Respiratory syncytial virus [335]
HIF prolyl hydroxylase inhibitors	HIF-1α stabilization	Inflammatory response [336]	Macrophage [337] Neutrophils [177] Dendritic cells [78] T cells [338]	SARS-CoV-2 [336] Influenza [339]
BAY87-2243/KC7F2	HIF-1α inhibition	Anti-inflammatory effects [22]	Dendritic cells [340] Macrophage [22] Neutrophils [214] Microglia [341]	SARS-CoV-2 [342] Respiratory syncytial virus [343]
15-Deoxy-Δ12,14-prostaglandin-j2	PPARy agonist	Induction of resolution profile [308]	Macrophage [307,308,309] Neutrophils [308] Dendritic cells [344]	Influenza [345] Respiratory syncytial virus [346]

## 5. Conclusions

During respiratory viral infections, several modifications in the infection sites occur and are responsible for the inflammatory microenvironment necessary for activation of the immune response against the invader. Innate and adaptive cells undergo metabolic reprogramming both to adapt to the inflammatory site and to activate an immune response against infection [28] (Figure 2). Exaggerated immune responses can cause significant collateral tissue damage, lung necrosis, fibrosis, and bronchiectasis, inducing adverse outcomes in patients [347]. Understanding which mechanisms trigger exaggerated immune responses is essential both to predict severe cases of respiratory infections and to develop therapeutic approaches to modulate the immune response and prevent the worsening disease. Hypoxia and HIF1α modulation can be interesting targets for the development of host-directed therapies for respiratory viruses.

## Figures and Tables

**Figure 1 viruses-15-00525-f001:**
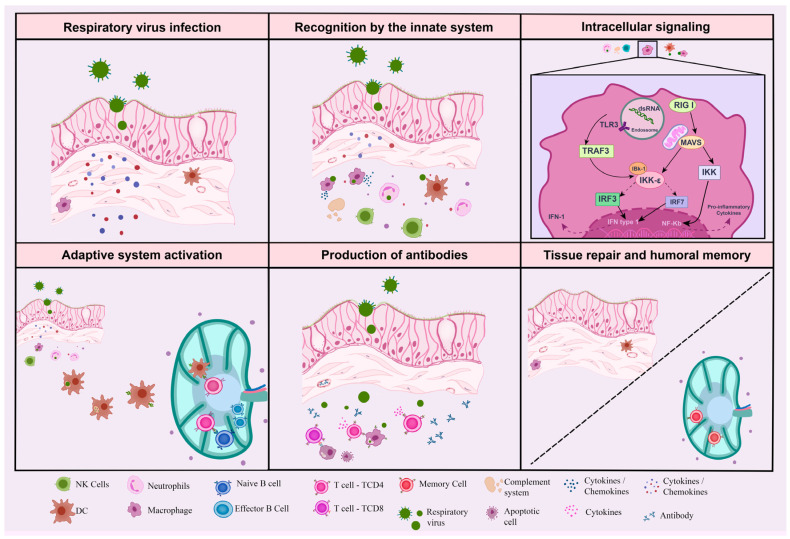
**The roles of innate and adaptative cells in respiratory viral infection**. Epithelial cells are barriers against respiratory viruses. The ciliary movement and mucus, together with tight junctions (TJs), form a mechanical, biological, and chemical barrier, regulating the innate and adaptative immune response by producing IFNs, nitric oxide (NO), cytokines, and chemokines that recruit and activate immune cells. When the barrier is broken and the virus enters, innate and adaptative immunity act at different time points to limit disease. Upon infection, innate immune cells, mainly macrophages and APCs, recognize pathogen-associated molecular patterns (PAMPs) and activate pattern-recognition receptors (i.e., TLRs), RIG-1-like receptors (RLRs), and cytosolic sensors (cGAS) that dimerize with their adaptor molecules and activate IRF3 and NF-κB, promoting cytokine release. The adaptive response occurs through lymphocyte cells: mature cDCs carrying virus-derived antigens migrate to lymphoid tissue, which are presented to B cells in the lymph nodes via MHC, activating a program of differentiation and activation, yielding the migration to the edge of the lymphoid follicle and proliferation. The virus-specific B cell response is generated largely in lymphoid tissues, either in regional lymph nodes or in mucosa-associated lymphoid tissues. T cells migrate to the infection site to exert their cytotoxic activity, while B cells activate and produce antibodies, coordinating virus-infected cell elimination. After infection resolution, memory T cells remain in the organism to activate a rapid response to a second encounter with the same antigen.

**Figure 2 viruses-15-00525-f002:**
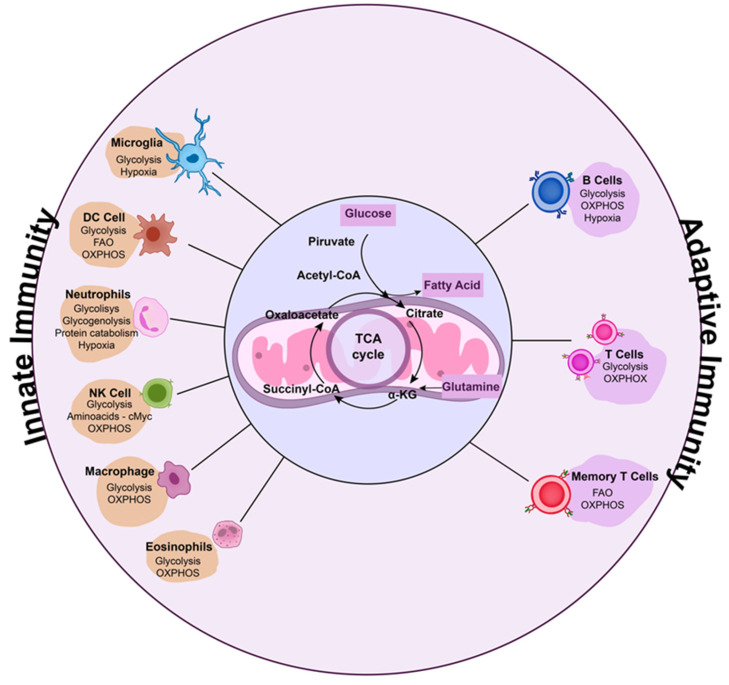
**The metabolic signature of immune cells upon viral infection**. Increased glycolytic rates typically yield effector phenotypes and both adaptative and innate immune cells rely on aerobic glycolysis, also known as the Warburg effect, to mount their pro-inflammatory response: increased glycolytic flux enhances the production of ATP via lactate dehydrogenase, producing lactate; OXPHOS is downregulated so that TCA intermediates can be used as signaling molecules and building blocks for other molecules, an event responsible for the known deviated TCA cycle (or ‘broken’ cycle), and electrons escape from the mitochondrial complexes, thereby generating mitochondrial ROS; higher glycolytic flux is also responsible for substrate supply for the PPP pathway, which produces NADPH for non-mitochondrial ROS; these events lead to the stabilization of the transcription factor HIF-1α that in turn activates the NF-κB pathway responsible for the production of cytokines, chemoattractants, biosynthetic molecules, and antiviral molecules. Glycolytic activation of cDCs allows the translocation of the glycolytic enzyme HK2 to the mitochondria to trigger the upregulation of costimulatory molecules, cytokine production, and enhanced T cell stimulatory capacity. In macrophages flux through glycolysis and the glycerol–phosphate and malate–aspartate shuttle increase, supporting energy metabolism and cytoplasmatic NADH production to support the production of pro-inflammatory metabolites and redox signals. Microglia activation is associated with enhanced aerobic glycolysis for ATP generation, increased expression of GLUT1 and hexokinase-2 to ROS, and NO production, as their generation depends on NADPH. Neutrophils rely on glycogenolysis to generate energy, and glucose diverts from glycolysis to PPP to generate NADPH and induces NADPH oxidase for ROS generation, essential to neutrophil extracellular trap (NET) formation and respiratory bursts. In NK cells, glycolysis and oxidative phosphorylation increase for IFNγ production and cytotoxic function: glucose is necessary for both glycolysis and OXPHOS, and mTORC1 is robustly regulated in activated NK cells, upregulating glucose uptake and glycolysis yielding the production of IFNγ and granzyme B. T effector cells are known to induce aerobic glycolysis upon viral infection, whereas memory T cells rely on mitochondrial metabolism and lipid oxidation, but still can rapidly shift to glycolysis. On the other hand, regulatory T cells instead prioritize mitochondrial lipid, pyruvate, and lactate oxidation. B cell activation and cytokine production enhance the glycolytic flux, while flux through the pentose phosphate pathway (PPP) is limited. Fatty acid oxidation is the main substrate to fuel OXPHOS in memory T cells.

## Data Availability

Not applicable.

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
