# Peer review of "Immunometabolic Signature during Respiratory Viral Infection: A Potential Target for Host-Directed Therapies"

_viruses, 2023, doi:10.3390/v15020525_

Round 1
Reviewer 1 Report (Previous Reviewer 1)
I have now looked at the responses and they are acceptable. The authors should make sure all "cDC's" in the paper are changed to "pDCs" especially in the new sentences they have added.Author Response
I have now looked at the responses and they are acceptable. The authors should make sure all "cDC's" in the paper are changed to "pDCs" especially in the new sentences they have added.
Answer: We appreciate the review suggestion. However, our aim in the topic on dendritic cells was to differentiate the type of response elicited by conventional DC (cDCs) and plasmacytoid DCs (pDCs) subtypes during viral infection in the lungs (lines 131-158). The types pDCs and cDCS originate from common DC progenitor (CDP), and diverge during their differentiation with the appearance of pre-DC and pDC precursors. Although cDCs are more frequent in the lungs in the absence of antigens, during viral infection, pDCs can be rapidly recruited and play an important role in the immune response to respiratory viruses. Therefore, we decided to describe how each type of DC is recruited and activated during viral infection (lines 159-181, cDCs; lines 224-242, pDCs). Finally, in lines 224-242, our aim was to describe and compare the metabolic changes that occur during the activation of the two types of DCs. According to the literature used and cited in the review, we correctly describe the classification of dendritic cells on cDCs and pDCs.
Reviewer 2 Report (Previous Reviewer 2)
The author has revised it according to the reviewer's request, and I agree to accept the manuscript.
Author Response
.
This manuscript is a resubmission of an earlier submission. The following is a list of the peer review reports and author responses from that submission.
Round 1
Reviewer 1 Report
This review compiles information from a large body of published literature but does not synthesize the knowledge to discuss immunometabolic signatures as potential targets for therapies. This point is only briefly stated at the end and left for the reader to further develop.
Minor Comments.
Specific comments regarding the figures
The legends on the two figures included in this text are excellent summaries of the preceding article information but are only broadly related to the figures they follow:
Figure 1 contains the basic progression of a respiratory virus infection flowing from infection, to innate immune response, to adaptive immune response, to tissue repair and humoral memory. The legend for figure 1 contains a large number of details about the figure that are not seen in the image such as PAMPs, TLRs, or adapter molecules, and fails to identify some of the larger details such as the draining lymph node which is prominent in the figure but not detailed in the image legend of the figure.
Figure 2 summarizes the major effectors of the cell types (glycolysis, OXPHOS, hypoxia, ect.) which were discussed briefly in the preceding text without indicating the direction that these metabolites change in response to viral infections. In the center of the figure is a foreshortened version of the TCA (Krebs) cycle which most of the included cells rely upon for proinflammatory responses. The figure would probably be better served by a table cataloging the information from the text rather than using a diagram that is not explained in the legend and which does not add to or diagram the conclusions of the article. While either the TCA or the immune cell descriptions might be useful on their own, their combination in this figure confounds the message of the image and following text.
Specific Comments Regarding the Text
The English used is correct and readable but I would suggest the following edits for clarity of the article:
· Line 146: cDCs is written, pDCs would make more sense in the sentence context
· Line 302: This information about M1 and M2-like macrophages would make more sense if it were near the previous information about M1/M2 at line 259
· Line 335: part of this sentence was duplicated within the sentence
· Line 386 and 387 seem to contradict each other, please clarify
· Line 402: wording is unclear, suggest rewrite to “high metabolic-demand is required to…”
· Figure 2: cite the Warburg effect mentioned in this figure legend but no where else in the article text
· Line 736: wording is unclear, suggest rewrite
Author Response
General Comment. The legends on the two figures included in this text are excellent summaries of the preceding article information but are only broadly related to the figures they follow:
Figure 1 contains the basic progression of a respiratory virus infection flowing from infection, to innate immune response, to adaptive immune response, to tissue repair and humoral memory. The legend for figure 1 contains a large number of details about the figure that are not seen in the image such as PAMPs, TLRs, or adapter molecules, and fails to identify some of the larger details such as the draining lymph node which is prominent in the figure but not detailed in the image legend of the figure.
Answer: We appreciate the review comment on figure 1 and made the changes suggested in the figure and in the legend (highlighted in yellow).
Lines 116-122: “The adaptive response occurs through lymphocyte cells: mature cDCs carrying virus-derived antigens migrate to lymphoid tissue, which are presented to B cells in the lymph nodes via MHC, activating a program of differentiation and activation, yielding the migration to the edge of the lymphoid follicle and proliferation. Virus-specific B cell response is generated largely in lymphoid tissues, either in regional lymph nodes or in mucosa-associated lymphoid tissues. T cells migrate to the infection site to exert their cytotoxic activity, while B cells activate and produce antibodies, coordinating virus-infected cell elimination”.
Figure 2 summarizes the major effectors of the cell types (glycolysis, OXPHOS, hypoxia, ect.) which were discussed briefly in the preceding text without indicating the direction that these metabolites change in response to viral infections. In the center of the figure is a foreshortened version of the TCA (Krebs) cycle which most of the included cells rely upon for proinflammatory responses. The figure would probably be better served by a table cataloging the information from the text rather than using a diagram that is not explained in the legend and which does not add to or diagram the conclusions of the article. While either the TCA or the immune cell descriptions might be useful on their own, their combination in this figure confounds the message of the image and following text.
Answer: We thank the reviewer for the comments. Indeed, the legend was not aligned with the figure 2 and we made the necessary changes, highlighted also in yellow.
Lines 809-827: “Glycolytic activation of cDCs allows the translocation of the glycolytic enzyme HK2 to the mitochondria to trigger the upregulation of costimulatory molecules, cytokine production and enhance T cell stimulatory capacity. In macrophages flux through glycolysis and the glycerol-phosphate and malate-aspartate shuttle increase, supporting energy metabolism and cytoplasmatic NADH production to support the production of pro-inflammatory metabolites and redox signals. Microglia activation is associated with enhanced aerobic glycolysis for ATP generation, increased expression of GLUT1 and hexokinase-2 to ROS, and NO production, as their generation depends on NADPH. Neutrophils rely on glycogenolysis to generate energy, and glucose diverts from glycolysis to PPP to generate NADPH and induces NADPH oxidase for ROS generation, essential to neutrophil extracellular traps (NET) formation and respiratory burst. In NK cells glycolysis and oxidative phosphorylation increase for IFNγ production and cytotoxic function: glucose is necessary for both glycolysis and OXPHOS, and mTORC1 is robustly regulated in activated NK cells, upregulating glucose uptake and glycolysis yielding the production of IFNγ and granzyme B. T effector cells are known to induce aerobic glycolysis upon viral infection, whereas memory T cells rely on mitochondrial metabolism and lipid oxidation, but still can rapidly shift to glycolysis. On the other hand, regulatory T cells instead prioritize mitochondrial lipid, pyruvate, and lactate oxidation. B cell activation and cytokine production enhance the glycolytic flux, while flux through the pentose phosphate pathway (PPP) is limited. Fatty acid oxidation is the main substrate to fuel OXPHOS in memory T cells”
The English used is correct and readable but I would suggest the following edits for clarity of the article:
Minor point 1: Line 146: cDCs is written, pDCs would make more sense in the sentence context
Answer: We would like to thank the reviewer for the careful review. Corrections are highlighted in yellow.
Line 149:” During respiratory viral infections, pDCs develop a similar response, taking up…”
Minor point 2: Line 302: This information about M1 and M2-like macrophages would make more sense if it were near the previous information about M1/M2 at line 259
Answer: We understand the reviewer’s suggestion and would like to explain the idea of the authors: describing macrophage polarization is to elucidate how, in a generalistic manner, macrophages are classified on behalf of their polarization state (M1/M2). Starting at line 256, we describe the dichotomy (M1/M2) that is applied to all macrophages and their subsets, but does not reflect the actual spectrum of macrophage polarization states. As it goes along the text, beginning in line 280, the idea is to describe how lung macrophages, and their subtypes, are classified according to their polarization states within the lungs and how the lung microenvironment is capable of changing these states. Since the review is focused on respiratory viruses, we think that it is suitable for the review a brief introduction of macrophage general polarization to contextualize the topic and a more specific point of view to describe the polarization of lung residing macrophage.
Minor point 3: Line 335: part of this sentence was duplicated within the sentence
Answer: Duplicated sentence in line 386 was removed.
Minor point 4: Line 386 and 387 seem to contradict each other, please clarify
Answer: We would like to thank the reviewer for the careful review. Corrections are highlighted in yellow.
Minor point 5: Line 402: wording is unclear, suggest rewrite to “high metabolic-demand is required to…”
Answer: We would like to thank the reviewer for the careful review. Corrections are highlighted in yellow.
Lines 405-406 “A common feature of viral infection is the high metabolic demand required to maintain rapid viral replication, which they do by reprogramming host metabolism and impeding immune defense”
Minor point 6: Figure 2: cite the Warburg effect mentioned in this figure legend but no where else in the article text
Answer: Aerobic glycolysis is also called as the Warburg effect, which is now referred to at the beginning of the text highlighted in yellow.
Line 207: “…OXPHOS fueled by the β-oxidation of lipids to aerobic glycolysis (also known as Warburg effect)…”
Line 801: Legend Figure 2: “…rely on aerobic glycolysis, also known as the Warburg effect…”
Minor point 7: Line 736: wording is unclear, suggest rewrite
Answer: We would like to thank the reviewer for the careful review. Corrections are highlighted in yellow.
Lines 833-835: “During respiratory viral infections, several modifications in the infection sites occur and are responsible for the inflammatory microenvironment necessary to activation of the immune response against the invader”
Reviewer 2 Report
Review:
The aim of this study is to review and summarize the immune metabolism characteristics during respiratory virus infection, so as to conduct host-directed therapy. The article introduces innate immunity, adaptive immunity and host-directed therapy one by one, which is rich in content and logical. But there are still some problems that need to be fixed.
1. The innate immune system consists of various white blood cells, the complement system, and the inflammatory response. The author focuses on several types of leukocytes (dendritic cells, macrophages, etc.). I wonder if the current status of respiratory virus research is related to complement, which is also an integral part of innate immunity.
2. Adaptive immunity and innate immunity complement each other, and the effector molecules of adaptive immunity can greatly promote the innate immune response, such as antibodies can promote the phagocytosis ability of phagocytes, which is called conditioning phagocytosis. After describing innate immunity and adaptive immunity, did the author consider the effect of the link between the two on respiratory viruses?
3. The title of the authors indicates potential targets for host-directed therapy, but the fourth part only deals with immune metabolic reprogramming and hypoxemia, and whether there is some lack of evidence for host-directed therapy.
4. The author should pay attention to the spelling mistakes when writing, such as IFN-I in line 133 instead of INF-I. And so on, hope the author seriously revised.
5. It is suggested that the author can make a table to sort out and summarize the article. For example, after respiratory virus infection, the important functional characteristics of various immune cells are described.
Author Response
General Comment. The aim of this study is to review and summarize the immune metabolism characteristics during respiratory virus infection, so as to conduct host-directed therapy. The article introduces innate immunity, adaptive immunity and host-directed therapy one by one, which is rich in content and logical. But there are still some problems that need to be fixed.
Answer: We thank the reviewer a lot for the supportive comments.
- The innate immune system consists of various white blood cells, the complement system, and the inflammatory response. The author focuses on several types of leukocytes (dendritic cells, macrophages, etc.). I wonder if the current status of respiratory virus research is related to complement, which is also an integral part of innate immunity.
Answer: We would like to thank the reviewer for the very good suggestion. We added a topic about complement system in text highlighted in yellow.
Lines 617-658: “Complement system is formed by a complex network of proteins organized in a proteolytic cascade to enhance the function of antibodies and phagocytic cells in response to pathogens infection. Three different ways can activate the complement system: classical, lectin and alternative pathways[236].
Classical activation, known as antibody-dependent, occurs by the recognition of antigen:antibodies complex formed by C1q protein linked to surface proteins of pathogens and the Fc region of the IgG and IgM proteins isotypes[236]. Lectin pathway is activated by highly conserved carbohydrate structures of pathogens, called pathogen-associated molecular patterns (PAMPs), that are recognized by recognition pattern receptors (PRRs)[236]. Finally, the alternative pathway is spontaneous activated by hydrolysis, remaining basally active which allows a rapid response during pathogen infection[236]. The three pathways converge to the C3 convertase production that leads to generation of the classical inflammation by anaphylatoxins, membrane attack complex (MAC) and opsonin, the effectors of complement system. The functions of complement system consist of neutralization of pathogens by opsonization, lyse and death of pathogens and host cells infected by formation of MAC, regulation of inflammatory response and enhancement of the adaptive immune response[236], [237].
Complement system acts as a functional bridge between he innate and adaptive immune system[238]. Classical activation of complement system depends on neutralization of pathogen antigen by B cell-produced antibodies[237]. In addition, complement system induces DCs maturation to presentation of foreign pathogens and activation of T cells[239]. Complement receptor CD21 modulates B cell functions (such as antigen internalization and presentation) and complement regulator CD46 regulates T cell-mediated immunity[238].
Viral infection actives complement system by three activation pathways, leading to viral neutralization by production of proinflammatory anaphylatoxins and formation of the MAC to lyse of enveloped viruses or infected host cells[240]. Viral antigens processed by complement system also active adaptive response[241], enhancing Th1 response[242] and B-cell memory viability[243] and modulating Treg and Th17 responses[244]. Antiviral response triggered by the complement system is essential in the neutralization of the respiratory viruses, however, recent data have shown that its hyperactivation is behind severe cases of Sars-CoV-2[245]–[248] and Influenza[249], [250] virus infection.
In addition to its direct role in innate and adaptive immune response activation, the complement system also plays a role in the modulation of immunometabolism[251]. Liszewski and colleagues[252] demonstrated that CD46 acts as an intracellular nutrient sensing and upregulates glucose transporter GLUT1 and aminoacids transporter LAT1, fueling T cell activation[252]. CD46 and TCR costimulation induces mitochondrial production of ROS that stimulates secretion of bioactive IL-1β and promotion of T helper 1 differentiation[253]”.
- Adaptive immunity and innate immunity complement each other, and the effector molecules of adaptive immunity can greatly promote the innate immune response, such as antibodies can promote the phagocytosis ability of phagocytes, which is called conditioning phagocytosis. After describing innate immunity and adaptive immunity, did the author consider the effect of the link between the two on respiratory viruses?
Answer: We would like to thank the reviewer for the suggestion. We added a brief text about the interaction between the innate immune system and the adaptative immune system, highlighted in yellow in lines 661-667: “The adaptive response occurs through the interaction between innate immune cells (mainly DCs and macrophages) and the adaptative immune response, the T and B cells. cDCs carrying virus-derived antigens migrate to lymphoid tissue, which are presented to T cells in the lymph nodes via MHC-II, activating a program of differentiation and activation, promoting the migration to the edge of the lymphoid follicle and proliferation. T cells migrate to the infection site to exert their cytotoxic activity, while B cells activate and produce antibodies, coordinating virus-infected cell elimination.
Others informations about link between innate and adaptive immunity are pointed out throughout the text, highlighted in yellow:
Line 149: “During respiratory viral infections, pDCs develop a similar response, taking up and processing viral antigens and efficiently presenting them to CD8+ and CD4+ T cells, helping to orchestrate an adequate immune response against the pathogens”
Line 275: “However, the evolution of adaptive immunity enabled macrophages with functions associated with both T and B cell responses: along with the professional antigen presentation cells (APCs) DCs, they serve as a key group of non-professional APCs and regulate adaptive immunity”
Line 389: “Macrophages depletion following influenza infection is associated with increased CD8+ T-cell response, and depletion of macrophages after 48h led to impaired CD8+ T-cell response”.
Line 679: “CD4+ T cells are activated after recognizing virus-derived MHC class II-associated peptides on APCs that also express co-stimulatory molecules[127], whereas CD8+ T cells are activated in the lymph nodes and recruited to the site of infection, eliminating infected cells via their cytolytic activity”
- The title of the authors indicates potential targets for host-directed therapy, but the fourth part only deals with immune metabolic reprogramming and hypoxemia, and whether there is some lack of evidence for host-directed therapy.
Answer: We understand the reviewer’s concern and we reformulated the topic “Immunometabolism and host-directed therapies to respiratory viruses”, lines 745 to 798 are highlighted in yellow.
Lines 645-798: “Viruses, especially RNA respiratory viruses, present a high mutation rate which may contribute to the frequent occurrence of vaccine-induced immunity escape and requires frequent vaccines update [25]. Host-directed interventions based on the immune response have shown promise in the treatment of respiratory viral infections, but these strategies sometimes come with unpleasant side effects. Cytokines treatment (i.e: type I interferons) increase immune response, however, it may be associated with exaggerated responses, leading to severe cases of respiratory diseases, such as cytokine storm [23], [24] and dysregulation of neutrophils [201], [202] and B cells[288]. Despite that, immunosuppressants (i.e: corticosteroids) suppress the immune response, preventing exaggerated inflammation, but they may benefit viral replication and propagation[291].
Immunometabolism reprogramming is associated with the regulation of immune response during viral infection (Figure 2). Understanding the metabolic regulation of cells during the immune response can contribute to the development of host-directed interventions for pathogens infections[291]. Immunometabolism has been demonstrated as a target for host-directed therapies for fungal diseases[35], tuberculosis[292], [293]. Recently therapeutic interventions for respiratory viral infections focused on immunometabolism have also been proposed. Some drugs targeting immunometabolism regulation have shown promising effects in the control of severe cases of viral infections (Table 1).
Dimethyl fumarate and 4-octyl-itaconate, derivatives of itaconate, are anti-inflammatory drugs that inhibit NLRP3 inflammasomes [294]. Liao and colleagues[295] demonstrated that the anti-inflammatory effects of 4-octyl-itaconate are also associated with glycolysis reduction via GAPDH inhibition. These compounds attenuate the airway inflammatory response to Sars-CoV-2 virus infection [296] and suppress pulmonary inflammation and mortality in influenza infections [297].
Metformin, an antidiabetic drug, that inhibits mTOR and actives AMPK, promotes antifibrotic effects on the lungs [298], [299]. Kamyshnyi and colleagues demonstrated that, during Sars-CoV-2 infection, AMPK activation by metformin activates lymphocyte subpopulations reducing hyperinflammation and controlling cytokine storm [300]. PI3K/AKT/mTOR pathway, which coordinates metabolic reprogramming (i.e: uptake and utilization of nutrients, including glucose and glutamine) and activation of immune cells[301]. Inhibition of the PI3K/mTOR reverted the disbalance of glucose metabolism in severe cases of pediatric influenza[302].
Hypoxemia is a defining feature of acute distress syndrome (ARDS), associated with complications of pulmonary inflammation [303], [304]. Viral replication in lung epithelial cells impairs gas exchange, leading to hypoxemia, which stabilizes HIF1α in immune cells and reprograms their metabolism. HIF-1α induces heightened rates of glycolysis to produce lactate and ATP, diverting glycolysis to PPP-dependent NADPH and ROS production [305]. HIF-1α stabilization is necessary for immune responses against pathogens but is also associated with an exaggerated inflammatory response by inducing cytokine storm [23] and elevated chemotaxis of monocytes and neutrophils to the infection site [306]. Hypoxia can dysregulate the response of macrophages[23], neutrophils[215], and lymphocytes[307]. Hypoxia altered the bone marrow hematopoiesis with consequences for the phenotype and number of monocytes and led to the persistence of inflammation[209]. Treatment with mCSF-1 (macrophage growth factor colony-stimulating factor 1) drove inflammation resolutions [209]. Understanding the hypoxia and HIF1 α modulation in both mild and severe infections by respiratory viruses may allow the development of new therapeutic approaches for the most severe cases of respiratory syndromes.
Lung injuries are triggered in severe cases of viral infection by an exaggerated inflammatory response[308]. PPARy, a nuclear receptor essential to lipid metabolism, is involved in the shift from inflammatory to anti-inflammatory immune cells polarization, modulating neutrophil function and macrophage activation[309]. 15-Deoxy-Δ12,14-prostaglandin-j2, a PPARy agonist, reduces the number of immune infiltrates and induces a resolving immune response, reducing exaggerated inflammation. [310]–[312]”
- The author should pay attention to the spelling mistakes when writing, such as IFN-I in line 133 instead of INF-I. And so on, hope the author seriously revised.
Answer: We would like to thank the reviewer for the careful review. We revised the text and corrected spelling mistakes.
- It is suggested that the author can make a table to sort out and summarize the article. For example, after respiratory virus infection, the important functional characteristics of various immune cells are described.
Answer: We would like to thank the reviewer for the suggestion. Our focus in this review was describe the metabolic changes that immune cells undergo during the response to a viral infection and propose that these metabolic alterations be considered for the development of host-directed therapies. Therefore, we have improved the legend of Figure 2 to summarize to metabolic changes in each immune cell type (lines 809-827). We also added a table (line 830) describing some metabolic targets and their inhibitors/agonists considered potential targets in the development of host-directed therapies.